# 3D Pose Estimation for Object Detection in Remote Sensing Images

**DOI:** 10.3390/s20051240

**Published:** 2020-02-25

**Authors:** Jin Liu, Yongjian Gao

**Affiliations:** State Key Laboratory of Information Engineering in Surveying, Mapping, and Remote Sensing, Wuhan University, Wuhan 430079, China; gaoyongjian@whu.edu.cn

**Keywords:** object detection, remote sensing images, 3D pose

## Abstract

3D pose estimation is always an active but challenging task for object detection in remote sensing images. In this paper, we present a new algorithm for predicting an object’s 3D pose in remote sensing images, called Anchor Points Prediction (APP). Compared to previous methods, such as RoI Transform, our object results of the final output can obtain direction information. We predict the object’s multiple feature points based on the neural network to obtain the homograph transformation relationship between object coordinates and image coordinates. The resulting 3D pose can accurately describe the three-dimensional position and attitude of the object. At the same time, we redefine the method $IoU_{APP}$ for calculating the direction and posture of the object. We tested our algorithm on the HRSC2016 dataset and the DOTA dataset with accuracy rates of 0.863 and 0.701, respectively. The experimental results show that the accuracy of the APP algorithm is significantly improved. At the same time, the algorithm can achieve one-stage prediction, which makes the calculation process easier and more efficient.

## 1. Introduction

In recent years, with the deepening of research and the improvement of computing power, deep learning has become more and more widely used in various fields. At the same time, the object detection algorithm has made great progress so far. In particular, remote sensing images has been a specific but active topic in computer vision [1,2]. Recent progresses in object detection in aerial images have benefited a lot from the R-CNN frameworks [1,3,4,5,6]. These methods use horizontal bounding boxes as the region of IoU and then rely on region-based features for category identification [2,7,8]. Faster-RCNN [4,5] leads to an elegant and effective solution where proposal computation is nearly cost-free given the detection network’s computation. A multi-stage object detection framework, the Cascade R-CNN, is proposed for the design of high-quality object detectors [6,9]. Additionally, FPN uses feature pyramids for object detection [10]; Yolt achieves object detection of high-resolution remote sensing images based on Yolo v3 [11,12]; and Yolo v3 is significantly faster than other methods in achieving the same accuracy [13]. These classic algorithms have different adaptation scenarios and greatly promote the development of this field. However, in remote sensing images, the object is often placed obliquely, so using an inclined box to detect the object will be more adaptive to the scene. These algorithms use a horizontal rectangular box to detect the object, so it does not accurately reflect the object pose of the remote sensing image to some extent. Also, these horizontal RoIs typically lead to misalignments between the bounding boxes and objects [8,14,15]. The RoI Transform algorithm locates the inclined box by predicting the rotation angle of the object box [8,16]. However, this algorithm has some problems. The first problem is that the rotation angle θ of the regression inclined box is ambiguous in most cases. This means that θ is equal to 0° and 180° corresponding to IoU is equal, but if the algorithm does not contain direction information, it will be considered the same type. The second problem is efficiency. It is a two-stage algorithm and the localization method relies entirely on Faster-RCNN [5,8]. The algorithm can only use the rectangle obtained by Faster-RCNN, and cannot use the feature information of the object region. CornerNet is a new one-stage approach to object detection by predicting the coordinates of the top-left and bottom-right points that does away with anchor boxes, which is more accurate and efficient [17,18,19,20]. However, predicting two points does not fully describe the information of the inclined box [5,21,22].

A 3D pose describes the three-dimensional pose of the camera relative to the object’s own coordinate system, not the pose of the object relative to the ground plane. Whether the reference object has z-coordinates and whether 3D information can be estimated are not the same thing. The object is on a plane and does not affect the rotation and translation of the camera observing the object relative to the three axes in three dimensions. Therefore, even objects on a two-dimensional plane will be observed in three-dimensional space out of the plane, which also has the 3D pose problem.

To solve the above problems, we propose the Anchor Points Prediction (called APP) algorithm. Different from other methods, we predict the position and attitude of the object by at least four corner points through the full convolution network, and can obtain the 3D pose by decomposing the homograph transformation matrix, and the algorithm is more efficient. The corner pooling layer used in the algorithm greatly improves the points prediction accuracy [17].

We give the correspondence between the predicted points and the available object information in Table 1, and a comparison of the traditional method and our method is shown in Figure 1. We have reason to believe that object detection by point prediction will become a new trend in the future.

## 2. Object Detection Based on APP

Any object detection problems can be attributed to the prediction of key points. The traditional rectangle object detection methods can be attributed to the prediction of two key points, such as Faster-RCNN, YOLO, and SSD (the upper-left and lower-right points of the rectangle) [5,21,24,25]; the 3D pose of the general object can be attributed to the prediction of eight points; the human body posture OpenPose can be attributed to the prediction of 18 key points of the human body [26]. We attribute the predicted inclined object box to a prediction of four points. The traditional inclined box detection methods have no direction information and may result in high accuracy. In addition, as shown in Figure 2, the boxes of the two objects whose center points are close but opposite in direction may cause the object to be lost in the NMS operation. The full name of NMS is non-maximum supply [27]. This method is used to search the local maximum and suppress the maximum. The purpose of this method is to eliminate redundant frames and find the best location for object detection.

Unlike traditional methods, we define a new calculation method that takes into account the overlap rate and direction consistency between the tilted boxes. Assuming there are two sets of object feature points, $\left\{P_{11}\ldots P_{1n}\right\}$, $\left\{P_{21}\ldots P_{2n}\right\}$, we define the $IoU_{APP}$ calculation formula between the two sets of points as follows:(1)$${\mathrm{IoU}}_{APP}=\frac{1}{1+\alpha \sqrt{\frac{2d_{12}}{d_{1}+d_{2}}}}.$$
We use $d_{12}=\sum_{i=1}^{n}{\left(P_{1i}-P_{2i}\right)}^{2}$, $d_{1}=\sum_{i=1}^{n}{\left(P_{1i}-\bar{P_{1}}\right)}^{2}$ and $d_{2}=\sum_{i=1}^{n}{\left(P_{2i}-\bar{P_{2}}\right)}^{2}$. $\bar{P_{1}}$ and $\bar{P_{2}}$ are the coordinates of the center point of point set 1 and point set 2, respectively. The definition mainly considers the deviation of the coordinate offset of the corresponding point from the size of the object itself. This deviation is relative. The larger the deviation, the smaller the $IoU_{APP}$. It is clear that the range of values of $IoU_{APP}$ is the same as the original IoU definition, which is [0, 1]. The larger the $IoU_{APP}$, the closer the two object cells are, and the $IoU_{APP}$ is equal to 1 when the two object cells are completely identical; the $IoU_{APP}$ is infinitely close to 0 when the two object cells are very different.

The two sets of the object feature points may be the two prediction units to be combined in the object detection NMS process, or may be the similarity calculation between ground truth and the predicted values.

According to Figure 3, we can get the calculation formulas for $IoU_{APP}$, IoU, and $IoU_{RBox}$ as follows, and we can get the relationship curve as shown in Figure 4.
(2)$$\begin{array}{cc} \; & {\mathrm{IoU}}_{APP}=\frac{1}{1+8\alpha \sin^{2}\frac{\theta }{2}},\alpha =1,2,3,\ldots \\ \; & \mathrm{IoU}=\frac{1}{{\left(1+2\frac{\left|\cos\theta \sin\theta \right|}{1+|\cos\theta |+|\sin\theta |}\right)}^{2}}\\ \; & {\mathrm{IoU}}_{rbox}=\frac{1}{\left|\cos\theta |+|\sin\theta \right|}. \end{array}$$

It can be seen from the above figure that when the object position scale is constant, only the $IoU_{APP}$ is significantly affected by the object direction angle θ, so only the $IoU_{APP}$ can describe the accuracy of the object direction angle. The IoU and $IoU_{RBox}$ are not affected by θ and cannot describe the accuracy of the object direction angle.

The mAP (mean Average Precision) obtained from the experimental data of RoI Transform is based on $IoU_{RBox}$ [8]. The mAP, which is used to evaluate the accuracy of object detection methods, is based on IoU between prediction boxes and ground truth boxes [28]. As can be seen from the above Figure, as long as the neural network that can recommend the horizontal box aligns with the center point, the $IoU_{RBox}$ is always greater than 0.5. That is, even if the direction angle prediction is wrong (predicted to be any angle from 0° to 360°), it is also hit when counting mAP, so the resulting mAP is virtually high. Thus, we proposed the solution $IoU_{APP}$. The $IoU_{APP}$ uses the method of regression coordinates to detect the object, and the method of evaluating mAP is more reasonable.

## 3. Anchor Points Prediction Algorithm

### 3.1. Neural Network Design

To predict the inclined box of the remote sensing object, we built a full convolutional network that predicts three scales in three different layers, each scale being the output by the APP of three different anchors’ array. Different anchors are used to detect objects with different aspect ratios in the image domain, as shown in Figure 5.

Our custom region layer is used to output the relative coordinates, the categories, and the information about whether the object exists or not. In the region layer, we used yolov3’s definition of anchors to implement n-weight anchor predictions based on the width and height of the object on the imaging surface. Each anchor represents a specific 2D wide height object. Following this concept, this particular 2D width and height corresponds to different APP range distributions. Each cell of the output array contains n×(4+1+c+2×4) output neurons. The meaning of the parameters are shown in Table 2.

We define the offset coordinate of point i in the range of a specific region as $\left(p_{w}\times \Delta x_{i},p_{h}\times \Delta y_{i}\right)$. As shown in Figure 6, $p_{w}$ and $p_{h}$ are the width and height of a particular anchor. We can use Equation (Equation 3) to calculate $P_{i}$.
(3)$$\left\{ \begin{array}{l} u_{i}=C_{x}+\frac{1}{2}+p_{w}x_{i}\\ v_{i}=C_{y}+\frac{1}{2}+p_{h}y_{i} \end{array}\right.$$

Then, the actual pixel offset coordinate of point i relative to the anchor boxes is $P_{i}=\left(u_{i},v_{i}\right)$. In addition, we define the loss function in the training procedure as follows, and we give the meaning of each parameter in Table 3.
(4)$$Loss(OutputLayer)=\{\begin{array}{l} \lambda_{nobj}{\left(0-Obj\right)}^{2}\;\mathrm{No}\;\mathrm{object}\;\mathrm{in}\;\mathrm{local}\;\mathrm{scope}\\ \lambda_{obj}{\left(1-Obj\right)}^{2}+\lambda_{pts}\sum_{i=1}^{4}{\left(P_{i}-{\bar{P}}_{i}\right)}^{2}+\lambda_{ROI}Loss_{ROI}\;\mathrm{object}\;\mathrm{within}\;a\;\mathrm{local}\;\mathrm{scope} \end{array}$$

We are more focused on the positioning learning of the inclined box determined by APP, so we reduce the weight of the horizontal box. Then, we use $\lambda_{ROI}=0.01$ to assist learning, and we can even set it to 0 to ignore the weight of the horizontal box. $Loss_{ROI}$ is the regression error of the center point and width of the object RoI, following yolo v2 [12].

The principle of judging whether there is an object in a local range is to calculate the maximum IoU between the default anchors’ boxes and all the ground truth boxes in that range. If the IoU exceeds a threshold, there will be an object. The principle of discrimination here is consistent with the processing of yolo v2 and yolo v3 [12,13]. Therefore, the sum of the differences in the three-layer APP prediction result and the ground truth is
(5)$$Loss=\sum_{\mathrm{output}\;\mathrm{Layer}}\;\mathrm{Loss}\;(\mathrm{Output}\;\mathrm{Layer}).$$

### 3.2. Training Procedure

Training datasets. We experimented with the DOTA dataset. The original DOTA images are high-resolution remote sensing images, which is not convenient for direct processing using the neural network. Therefore, first of all, the raw data needs to be standardized. The method we took was to randomly select an image point, then center the point, and align the center point (W/2, H/2) of the transformed image (W, H) for random affine transformation. The scale of the transformation is 0.5 to 1.5 times the original image. Then, we obtain the sample images.

Training and testing. At training, we used 80% DOTA images, and all the processed images were resized into 416*416 and sent to the neural network. After training, the remaining 20% of images were used for testing. In the training process, the choice of multiple anchors followed the strategy of yolo v3, and the process of backpropagation of the loss layer was divided into two phases.

Phase 1: Scan each output of the output layer array. According to the ground truth set and the boxes determined by predicted APP coordinates, the output region can be obtained from the maximum IoU between them. If $IoU_{max}$ is less than ε, the corresponding object presence expectation output value will be set to 0 for backpropagation correction.

Phase 2: Scan the rows and columns of each GTBox, and correct the largest anchor of the IoU between the default rectangle of n anchors at this position and the GTBox. Then, set the expected value of the object field to 1, and the loss of APP is calculated according to Equation (Equation 19), and the expected value of the softmax segment is set to perform backpropagation correction.

Application. According to the four-sided output bounding box surrounded by four APP coordinates, the four points of the inclined box are further obtained by Equation (Equation 19), and the point coordinates are converted to the large image according to Equation (Equation 5).

### 3.3. Calculation of the Object of 3D Pose

The conventional methods often calculate the 3D pose of the object by matching the local features extracted in the 2D image with the features in the object 3D model to be detected, but these methods are not accurate enough [29]. Therefore, based on the key point coordinates of the object output from the region layer, we use the perspective transformation method to calculate the 3D pose. Figure 7 shows the computational process of the objects’ 3D pose. We use two methods, PnP [30] and homograph. The following describes these two methods.

#### 3.3.1. PnP Method

As shown in Figure 7, we can get the coordinates$\left\{P_{1},P_{2},P_{3},P_{4}\right\}$ of the four feature points of the object in the training part. These feature point coordinates are relative to the coordinates of the complete satellite image. If the image is cropped and resized, these coordinates need to be transformed to the coordinate system of the original satellite image. The inference part obtains the 3D pose of the object based on the correspondence between the four feature points and the body coordinates of the four or eight corner points of the object. Assume that the length and width of the bounding box of the object is *W* and *H*, and the height of the object from the ground is $H_{g}$. We define the eight points of the bounding box of the object’s own coordinate system as:(6)$$\left\{\begin{array}{c} \left[-\frac{W}{2},-\frac{H}{2},\pm \frac{H_{g}}{2}\right],\left[\frac{W}{2},-\frac{H}{2},\pm \frac{H_{g}}{2}\right]\\ \left[-\frac{W}{2},\frac{H}{2},\pm \frac{H_{g}}{2}\right],\left[\frac{W}{2},\frac{H}{2},\pm \frac{H_{g}}{2}\right] \end{array}\right.$$
Since the distance from the camera to the ground object is much larger than the object’s own height $H_{g}$, the image coordinates of the two feature points at different heights of the object at the same latitude and longitude are very close on the image. Therefore, the corresponding relationship between the eight points of the object’s bounding box and the key points $\left\{P_{1},P_{2},P_{3},P_{4}\right\}$ of the image coordinates is:(7)$$\left\{\begin{array}{cc} \left[-\frac{W}{2},-\frac{H}{2},\pm \frac{H_{g}}{2}\right]----P_{1} & \;\mathrm{left}\;\mathrm{top}\;\\ \left[\frac{W}{2},-\frac{H}{2},\pm \frac{H_{g}}{2}\right]----P_{2} & \;\mathrm{right}\;\mathrm{top}\;\\ \left[-\frac{W}{2},\frac{H}{2},\pm \frac{H_{g}}{2}\right]----P_{3} & \;\mathrm{right}\;\mathrm{bottom}\;\\ \left[\frac{W}{2},\frac{H}{2},\pm \frac{H_{g}}{2}\right]----P_{4} & \;\mathrm{left}\;\mathrm{bottom}\; \end{array}\right.$$
According to this correspondence, we call the PnpSolve function in OpenCV to solve the camera external parameters R and T, where R is the attitude matrix of the camera relative to the object, and T is the position of the object relative to the camera.

#### 3.3.2. Homograph Method

We define the object as an inclined box with a width W and a length H. The origin of the object’s coordinate system is defined at the center of the box. Considering the particularity of remote sensing images, the four vertices of the object in the object coordinate system are similar on a ground plane π, and the four vertices of the object in the object coordinate system are marked as:(8)$$\left\{\begin{array}{c} \left[-\frac{W}{2},-\frac{H}{2}\right],\left[\frac{W}{2},-\frac{H}{2}\right]\\ \left[\frac{W}{2},\frac{H}{2}\right],\left[-\frac{W}{2},\frac{H}{2}\right] \end{array}\right.$$

According to the DOTA data format, the four points are sorted clockwise from the upper left corner. Assuming that the aspect ratio of the object is also unknown, set to α, then the four object points can be written as:(9)$$\left\{\begin{array}{c} \left[-\frac{W}{2},-\frac{\alpha W}{2}\right],\left[\frac{W}{2},-\frac{\alpha W}{2}\right]\\ \left[\frac{W}{2},\frac{\alpha W}{2}\right],\left[-\frac{W}{2},\frac{\alpha W}{2}\right] \end{array}\right.$$

According to the principle of satellite remote sensing imaging, the line of sight of the imaging camera of the remote sensing image is perpendicular to the ground plane π, so the conversion of the four points on the object plane to the image plane follows the rotation transformation matrix:(10)$$\left[\begin{array}{c} u\\ v \end{array}\right]=\left[\begin{array}{ccc} a_{11} & a_{12} & a_{13}\\ -a_{12} & a_{11} & a_{23} \end{array}\right]\left[\begin{array}{c} X\\ Y\\ 1 \end{array}\right]$$

Conversely, the conversion of four points on the image plane to the four points on the object plane follows the inverse of Equation (Equation 8):(11)$$\left[\begin{array}{c} X\\ Y \end{array}\right]=\left[\begin{array}{ccc} a_{11}^{\prime } & a_{12}^{\prime } & a_{13}^{\prime }\\ -a_{12}^{\prime } & a_{11}^{\prime } & a_{23}^{\prime } \end{array}\right]\left[\begin{array}{c} u\\ v\\ 1 \end{array}\right]$$
According to the basic principle of affine transformation,
(12)$$\begin{array}{cc} \left[\begin{array}{ccc} a_{11} & a_{12} & a_{13}\\ -a_{12} & a_{11} & a_{23} \end{array}\right] & ={\left[\begin{array}{ccc} a_{11}^{\prime } & a_{12}^{\prime } & a_{13}^{\prime }\\ -a_{12}^{\prime } & a_{11}^{\prime } & a_{23}^{\prime } \end{array}\right]}^{-1}\\ \; & =\frac{1}{{a_{11}^{\prime }}^{2}+{a_{12}^{\prime }}^{2}}\left[\begin{array}{ccc} a_{11}^{\prime } & -a_{12}^{\prime } & a_{12}^{\prime }a_{23}^{\prime }-a_{13}^{\prime }a_{11}^{\prime }\\ a_{12}^{\prime } & a_{11}^{\prime } & -\left(a_{11}^{\prime }a_{23}^{\prime }+a_{13}^{\prime }a_{12}^{\prime }\right) \end{array}\right]. \end{array}$$
In order to solve the four parameters ${a_{11}}^{\prime }$, ${a_{12}}^{\prime }$, ${a_{13}}^{\prime }$, ${a_{23}}^{\prime }$ of affine transformation, we obtain the formulas of the four parameters and α according to Equation (Equation 7):(13)$$\left[\begin{array}{ccccc} u & v & 1 & 0 & 0\\ v & -u & 0 & 1 & -Y \end{array}\right]\left[\begin{array}{c} a_{11}^{\prime }\\ a_{12}^{\prime }\\ a_{13}^{\prime }\\ a_{23}^{\prime }\\ \alpha \end{array}\right]=\left[\begin{array}{c} X\\ 0 \end{array}\right].$$
The four-point coordinates in Equation (Equation 9) are sequentially brought into X and Y in the above equation,
(14)$$\left\{\begin{array}{cc} \left[X_{1},Y_{1}\right] & =\left[-\frac{W}{2},-\frac{\alpha W}{2}\right]\\ \left[X_{2},Y_{2}\right] & =\left[\frac{W}{2},-\frac{\alpha W}{2}\right]\\ \left[X_{3},Y_{3}\right] & =\left[\frac{W}{2},\frac{\alpha W}{2}\right]\\ \left[X_{4},Y_{4}\right] & =\left[-\frac{W}{2},\frac{\alpha W}{2}\right] \end{array}\right.$$
Then, we can get
(15)$$\left[\begin{array}{ccccc} u_{1} & v_{1} & 1 & 0 & 0\\ v_{1} & -u_{1} & 0 & 1 & -Y_{1}\\ u_{2} & v_{2} & 1 & 0 & 0\\ v_{2} & -u_{2} & 0 & 1 & -Y_{2}\\ u_{3} & v_{3} & 1 & 0 & 0\\ v_{3} & -u_{3} & 0 & 1 & -Y_{3}\\ u_{4} & v_{4} & 1 & 0 & 0\\ v_{4} & -u_{4} & 0 & 1 & -Y_{4} \end{array}\right]\left[\begin{array}{c} a_{11}^{\prime }\\ a_{12}^{\prime }\\ a_{13}^{\prime }\\ a_{23}^{\prime }\\ \alpha \end{array}\right]=\left[\begin{array}{c} X_{1}\\ 0\\ X_{2}\\ 0\\ X_{3}\\ 0\\ X_{4}\\ 0 \end{array}\right].$$
After solving Equation (Equation 14) by the least two multiplication method, it is brought into Equation (Equation 16) to get $a_{11}$, $a_{12}$, $a_{13}$, and $a_{23}$. According to the principle of perspective transformation, we can get
(16)$$\begin{array}{cc} K^{1}H={\left[\begin{array}{ccc} f & 0 & c_{x}\\ 0 & f & c_{y}\\ 0 & 0 & 1 \end{array}\right]}^{-1}\left[\begin{array}{ccc} a_{11} & a_{12} & a_{13}\\ a_{12} & a_{11} & a_{23}\\ 0 & 0 & 1 \end{array}\right] & =\left[\begin{array}{ccc} \frac{1}{f} & 0 & -\frac{c_{x}}{f}\\ 0 & \frac{1}{f} & -\frac{c_{y}}{f}\\ 0 & 0 & 1 \end{array}\right]\left[\begin{array}{ccc} a_{11} & a_{12} & a_{13}\\ a_{12} & a_{11} & a_{23}\\ 0 & 0 & 1 \end{array}\right]\\ \; & =\left[\begin{array}{ccc} \frac{a_{11}}{f} & \frac{a_{12}}{f} & \frac{a_{13}-c_{x}}{f}\\ -\frac{a_{12}}{f} & \frac{a_{11}}{f} & \frac{a_{23}-c_{y}}{f}\\ 0 & 0 & 1 \end{array}\right]. \end{array}$$
Columns 1, 2 of $K^{-1}H$ are then unitized to obtain the matrix:(17)$${\left[K^{-1}H\right]}_{1}=\frac{f}{\sqrt{a_{11}^{2}+a_{12}^{2}}}\left[\begin{array}{ccc} \frac{a_{11}}{f} & \frac{a_{12}}{f} & \frac{a_{13}-c_{x}}{f}\\ -\frac{a_{12}}{f} & \frac{a_{11}}{f} & \frac{a_{23}-c_{y}}{f}\\ 0 & 0 & 1 \end{array}\right].$$
The attitude matrix $R=[c_{1},c_{2},c_{1}\times c_{2}]$. The columns 1 and 2 of ${\left[K^{-1}H\right]}_{1}$ construct the columns 1, 2 $c_{1},c_{2}$ of the matrix R. The third column of ${\left[K^{-1}H\right]}_{1}=$ is the offset T of the object relative to the camera.

Based on the above, we summarize the calculation process of the 3D pose for remote sensing image objects.

(1)Predict the APP coordinates of each object through the neural network.(2)According to Equation (Equation 15), we can get the inverse affine transformation parameters ${a_{11}}^{\prime }$, ${a_{12}}^{\prime }$, ${a_{13}}^{\prime }$, ${a_{23}}^{\prime }$ and the object width-to-length ratio α.(3)According to Equation (Equation 12), we can get the affine transformation matrix from the object coordinate system to the image coordinate system:
(18)$$A=\left[\begin{array}{ccc} a_{11} & a_{12} & a_{13}\\ -a_{12} & a_{11} & a_{23} \end{array}\right]$$(4)Matrix ${\left[K^{-1}H\right]}_{1}$ is obtained from Equation (Equation 17).(5)Attitude matrix R and displacement T can be obtained by decomposing matrix ${\left[K^{-1}H\right]}_{1}$.

#### 3.3.3. Object Spatial Location Using Remote Sensing Image

As shown in Figure 8, the geometric transformation of the satellite relative to the earth $\left[\begin{array}{cc} R_{s} & t_{s}\\ & 1 \end{array}\right]$ can be obtained accurately; the rigid body connection transformation $\left[\begin{array}{cc} R_{c} & t_{c}\\ & 1 \end{array}\right]$ of the satellite camera relative to the satellite can also be measured. Then, through the method based on APP, the transformation of the object relative to the camera can be obtained as $\left[\begin{array}{cc} R^{T} & T\\ & 1 \end{array}\right]$. To sum up, it can be calculated that the 3D transform of the object relative to the earth is converted into $\left[\begin{array}{cc} R^{T} & T\\ & 1 \end{array}\right]\left[\begin{array}{cc} R_{c} & t_{c}\\ & 1 \end{array}\right]\left[\begin{array}{cc} R_{s} & t_{s}\\ & 1 \end{array}\right]$.

## 4. Experiments and Analysis

### 4.1. Experimental Details

Experimental data. We conducted experiments on the DOTA and HRSC2016 datasets. For the DOTA dataset, we cut the image into subgraphs with a resolution of 1024×1024. At training, we used batch-size = 64, and the learning rate was 0.0001 in the first 10,000 training sessions, and after every 20,000 increments, the learning rate was reduced by 0.1 times until the learning rate was equal to 0.000001. Our full convolutional network supports the input of images with different resolutions. We tested two different resolution inputs for 416×416 and 1024×1024. The 416×416 model can get an output of 13×13, 26×26, 52×52 scales, and the 1024×1024 model can get an output of 32×32, 64×64, and 128×128 scales, which correspond to three different object types of large, medium, and small scales, respectively. In the detecting, we first divided the DOTA image into blocks. The blocks needed to overlap to avoid cross-border loss of the object. Finally, we calculated and counted the mAP of the object. mAP is the mean Average Precision, which means the average of the AP of all object categories. Table 4 and Table 5 are the statistics on the HRSC2016 and DOTA data sets, respectively. As can be seen from these two tables, the APP algorithm has the highest mAP.

We also tested five sets of models with different parameters. According to Equation (Equation 8), we could get the IoU between the predicted object point set and the true value point set, and calculate the corresponding mAP of each of these models. Table 6 and Table 7 are model parameters and experimental results, respectively. By comparing the experimental results, we found that the mAP of the first set of model parameters was the highest.

The efficiency of the algorithm is one of the most important indicators for measuring the quality of the algorithm. Our model can input images in two different resolutions. As shown in Table 8, we used a one-stage process including NMS operation, so it was more efficient. Table 9 is the mean length of the objects. Assuming that the width W of all objects is equal to 1, the average length of each type of object also can be solved.

Experimental results. We improved yolo v3 and added four corner points of the APP prediction object based on the original region layer. The local object was predicted according to Equation (Equation 19), and then the coordinates were converted to the large image to obtain the homograph transform. Further, we decomposed the three-dimensional attitude R and displacement T of the object relative to the camera. Figure 9 is the result of the experiment. Figure 10 is an incorrectly labeled image that we can detect correctly after training.

For large images such as DOTA, we used the method of block synthesis. The large image is divided into a number of sub-blocks, each of which is just the standard size of the neural network input layer $\left[S_{w},S_{h}\right]$, and the neural network was used to detect the object APP coordinates $\left(u_{i},v_{i}\right)$ in each sub-block range. Assuming that the coordinate of the upper-left corner of a sub-block is (left,top), the coordinate of the point in the full image is
(19)$$\left\{\begin{array}{c} u_{ibig}=\mathrm{left}+u_{i}\\ v_{ibig}=\mathrm{top}+v_{i} \end{array}\right.$$

Converting all the objects of the sub-image to the large image, and in finally performing the NMS operation, in order to avoid the object being cut by the block, overlap between blocks was needed, and the overlap length was longer than the minimum object length. The overlap length of the horizontal and vertical sides was initially set at 20% of the basic length, as shown in Figure 11.

### 4.2. Error Analysis

We used the method of projecting pixel points to comprehensively evaluate the prediction accuracy of the 3D pose. The process is shown in Figure 7. When evaluating the accuracy of the 3D pose, the position and attitude of the object in three-dimensional space could be considered comprehensively by using the method of pixel projection error [36]. According to the nature of the attitude matrix R, the three rows of the matrix represent the three unit vectors of the x, y, and z axis of the camera coordinate system relative to the object, and then the three rows of R, in turn, represent the x, y, and z axis relative to the three unit vectors of the camera coordinate system. Since the y-axis of the object ontology coordinate system is the negative direction of the object orientation, the negative vector $-r_{12},-r_{22}$ of the second column of the matrix R describes the direction of the object. Therefore, the azimuth angle of the object relative to the camera coordinate system $\theta =\mathrm{artan}\left(-r_{12},-r_{22}\right)$. The coordinate calculation Equation is
(20)$$z_{i}\left[\begin{array}{c} u_{i}\\ v_{i}\\ 1 \end{array}\right]=KR\left(X_{i}-T\right)\;i=1,2,3,4,$$
where K is the internal parameter matrix of the camera, R and T are the attitude matrix and the displacement matrix obtained according to the algorithm of Section 4.1, and $X_{i}$ is the coordinates of the four projection points of the object box close to the ground. The pixel error is
(21)$$e^{2}={\left|\left[\begin{array}{c} u_{i}\\ v_{i}\\ 1 \end{array}\right]-\left[\begin{array}{c} {\bar{u}}_{i}\\ {\bar{v}}_{i}\\ 1 \end{array}\right]\right|}^{2}.$$
According to this formula, we could get the angular error and pixel error of the predicted object. We tested five different sets of parameters to get different models. It can be seen from Table 10 that the error of the first set of parameters is the smallest.

## 5. Conclusions

In this paper, we proposed a new Anchor Points Prediction algorithm that can accurately determine the position and attitude of the object’s three-dimensional space. Differing from the traditional methods of predicting object RoI or inclined box, we used the neural network to predict multiple feature points to detect the objects. This algorithm is a one-stage algorithm, and its accuracy and efficiency have been greatly improved. It not only uniquely determines the direction of the object, but also calculates the 3D pose of the object from the APP coordinates. We believe that the APP algorithm can be better applied to object detection. Moreover, the point prediction algorithm has broad application prospects and may become a new trend in the future. Our method also has some shortcomings. For a slender object like Harbor, the bounding box is relatively large, and the object occupies only a small part of the bounding box. In this way, the features in the extracted ROI region will be inaccurate, leading to a decrease in the accuracy of predicting key points of the object.

## Figures and Tables

**Figure 1 sensors-20-01240-f001:**
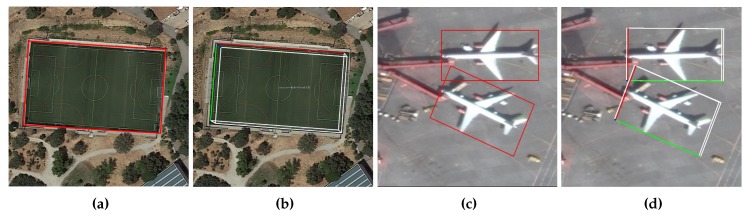
Object detection comparison. (**a**,**c**) Traditional inclined box. The inclined box has symmetry, so it is not possible to uniquely describe the direction of the object in the 2D image space, so the object has four possible directions. (**b**,**d**) The 3D pose diagram obtained from APP. The X-axis of the object is marked with red, the Y-axis is marked with green, and the Z-axis is marked with blue. The X-axis points to the right side of the object; the negative direction of Y-axis indicates the static direction of the object; and the Z-axis points to the ground.

**Figure 2 sensors-20-01240-f002:**
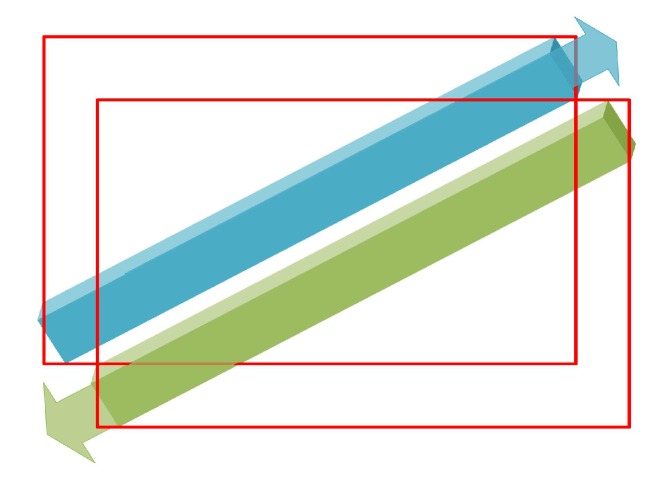
Two objects in opposite directions are prone to loss in NMS operations.

**Figure 3 sensors-20-01240-f003:**
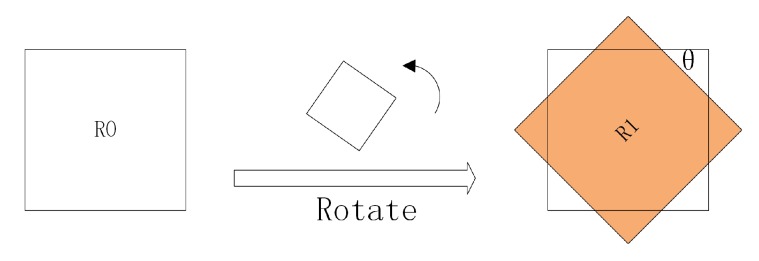
Two objects in opposite directions are prone to loss in NMS operations.

**Figure 4 sensors-20-01240-f004:**
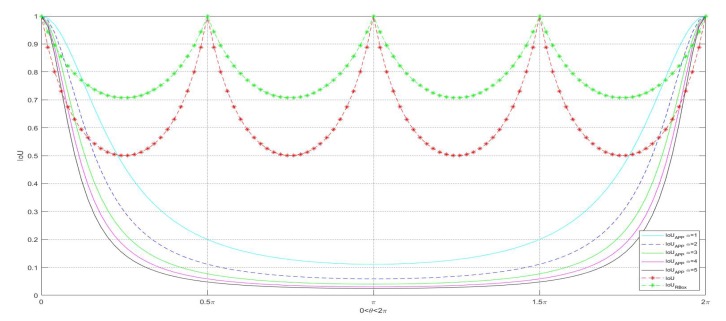
Relationship between $IoU_{APP}$, IoU, and $IoU_{RBox}$.

**Figure 5 sensors-20-01240-f005:**
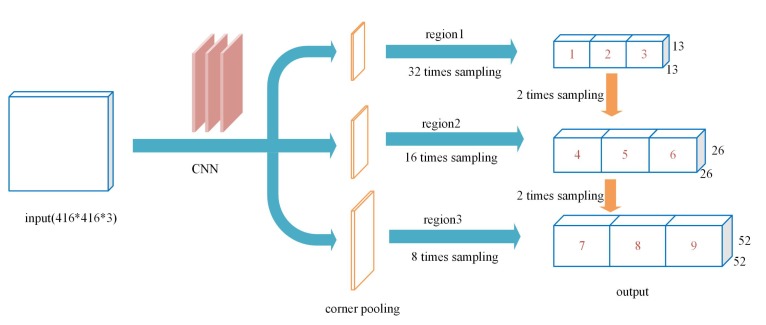
The architecture of the network.

**Figure 6 sensors-20-01240-f006:**
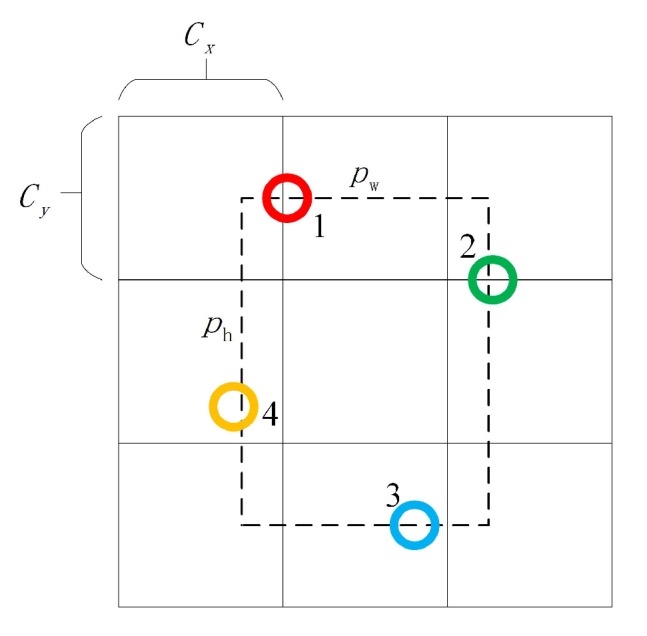
Anchor Points Prediction algorithm in the output grid.

**Figure 7 sensors-20-01240-f007:**
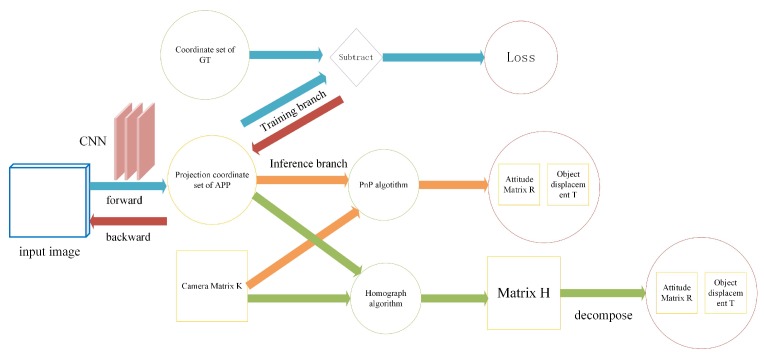
The schematic diagram of the 3D pose’s computational process.

**Figure 8 sensors-20-01240-f008:**
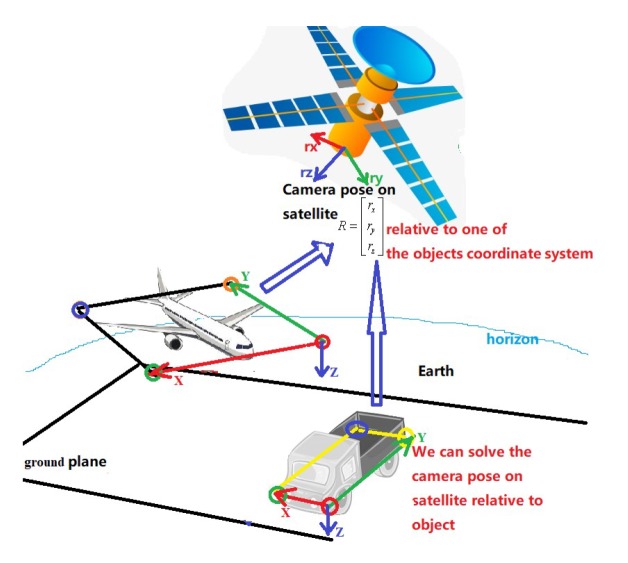
Application of remote sensing image for object spatial location.

**Figure 9 sensors-20-01240-f009:**
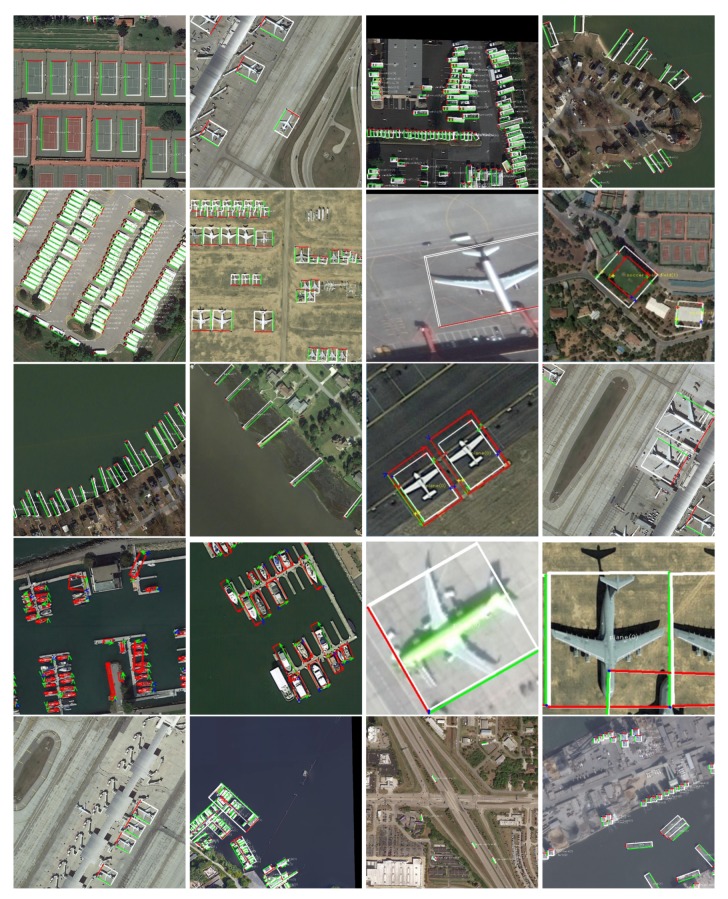
3D pose renderings. The red arrow direction and the green arrow direction in the figure are the X-axis and Y-axis of the object itself, respectively.

**Figure 10 sensors-20-01240-f010:**
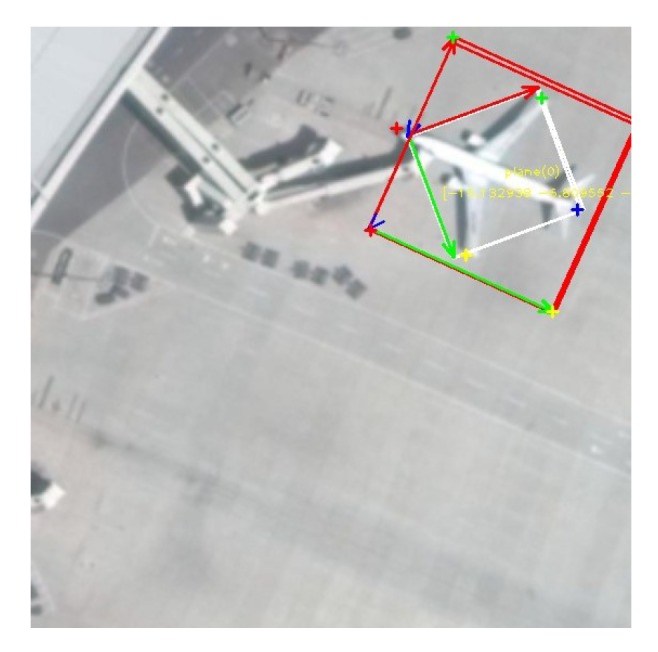
The inner box is not accurate in the DOTA1.0 training dataset, but we get a more accurate box by prediction.

**Figure 11 sensors-20-01240-f011:**
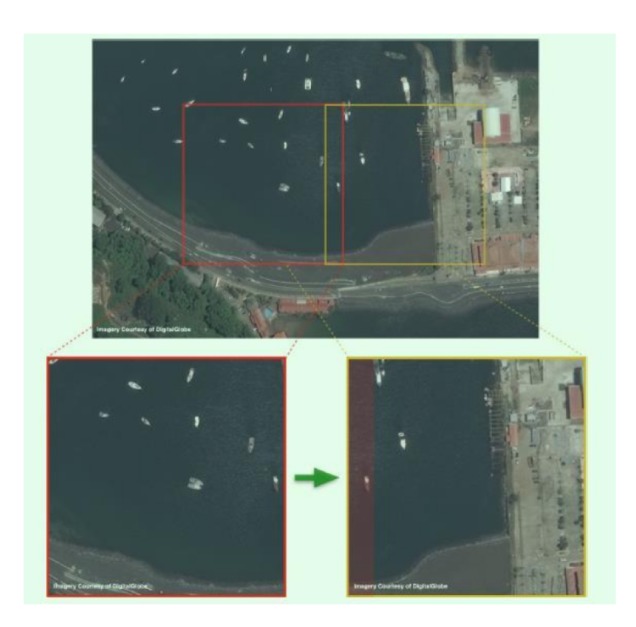
Large image segmentation.

**Table 1 sensors-20-01240-t001:** Correspondence between predicted points and available object information [17,23].

Prediction Points	Object Information
2	RoI
4	inclined box
≥4	3D pose

**Table 2 sensors-20-01240-t002:** The meaning of the parameters of the output.

Parameter	4	n	1	c	2×4
Description	bounding box coordinates	number of anchors	existing object	number of classes	$\left(\Delta x_{1},\Delta y_{1}\right)$ $\left(\Delta x_{2},\Delta y_{2}\right)$ $\left(\Delta x_{3},\Delta y_{3}\right)$ $\left(\Delta x_{4},\Delta y_{4}\right)$

**Table 3 sensors-20-01240-t003:** The meaning of the parameters of Loss function.

Parameter	$\lambda_{noobj}$	$\lambda_{obj}$	$\lambda_{ROI}$	$\lambda_{pts}$	$P_{i}$	$\bar{P_{i}}$
Description	loss weight of non-object	loss weight of object	loss weight of object RoI	loss weight of each point	predicted value of the point $\left(u_{i},v_{i}\right)$	ground truth of the point $\left(\bar{u_{l}},\bar{v_{l}}\right)$

**Table 4 sensors-20-01240-t004:** Comparisons with the state-of-the-art methods on HRSC2016 [8].

Method	CP [15]	BL2 [15]	RC1 [15]	RC2 [15]	$R^{2}\mathrm{CNN}$ [31]	RRD [32]	RoI Trans. [8]	APP
mAP	55.7	69.6	75.7	75.7	79.6	84.3	86.2	86.3

**Table 5 sensors-20-01240-t005:** Comparisons with state-of-the-art detectors on DOTA. There are 15 categories, including Baseball diamond (BD), Ground track field (GTF), Small vehicle (SV), Large vehicle (LV), Tennis court (TC), Basketball court (BC), Stor- age tank (ST), Soccer-ball field (SBF), Roundabout (RA), Swimming pool (SP), and Helicopter (HC) [8].

Method	Plane	BD	Bridge	GTF	SV	LV	Ship	TC	BC	ST	SBF	RA	Harbor	SP	HC	mAP
FR-O [33]	79.42	77.13	17.7	64.05	35.3	38.02	37.16	89.41	69.64	59.28	50.3	52.91	47.89	47.4	46.3	54.13
RRPN [34]	80.94	65.75	35.34	67.44	59.92	50.91	55.81	90.67	66.92	72.39	55.06	52.23	55.14	53.35	48.22	60.67
R2CNN [35]	88.52	71.2	31.66	59.3	51.85	56.19	57.25	90.81	72.84	67.38	56.69	52.84	53.08	51.94	53.58	61.01
DPSRP [8]	81.18	77.42	35.48	70.41	56.74	50.42	53.56	89.97	79.68	76.48	61.99	59.94	53.34	64.04	47.76	63.89
RoI Trans. [8]	88.53	77.91	37.63	74.08	66.53	62.97	66.57	90.5	79.46	76.75	59.04	56.73	62.54	61.29	55.56	67.74
PnP	87.98	75.38	45.93	71.26	65.10	68.93	77.04	87.63	81.59	78.96	58.54	57.20	63.95	62.32	49.01	68.72
APP	89.06	78.23	43.52	76.39	68.42	71.62	79.05	90.42	81.51	80.51	59.48	58.91	64.21	62.19	48.46	70.13

**Table 6 sensors-20-01240-t006:** Models with different parameters. We used five groups of loss weights to get different parameter models and compared their recognition accuracy.

Number	$\lambda_{noobj}$	$\lambda_{obj}$	$\lambda_{ROI}$	$\lambda_{pts}$	$\lambda_{softmax}$
1	1	2	0.0001	1	1
2	1	3	0.0001	1	1
3	1	5	0.0001	2	1
4	1	2	0.0001	1.5	1
5	1	2	0.1	1	1

**Table 7 sensors-20-01240-t007:** mAP for each model. It can be seen from the table that the model obtained by row 1 has the highest overall accuracy.

Method	Plane	BD	Bridge	GTF	SV	LV	Ship	TC	BC	ST	SBF	RA	Harbor	SP	HC	mAP
1	89.06	78.23	43.52	76.39	68.42	71.62	79.05	90.42	81.51	80.51	59.48	58.91	64.21	62.19	48.46	70.13
2	87.49	79.41	41.92	75.82	67.39	72.03	79.05	88.25	82.18	82.43	60.91	58.38	63.19	61.42	47.24	69.81
3	85.97	77.11	49.01	73.32	71.04	70.28	77.37	85.72	78.94	76.35	57.83	59.30	61.67	59.65	45.47	68.60
4	82.43	78.13	53.57	71.54	66.77	69.40	75.09	82.18	75.59	78.63	63.19	61.67	60.66	60.91	52.30	68.80
5	83.44	71.65	54.58	70.53	68.51	71.54	77.87	88.25	82.43	77.62	58.38	55.59	59.39	57.62	47.24	68.31

**Table 8 sensors-20-01240-t008:** Speed comparison with other methods (unit: ms). The LR-O, RoI Trans, and DPSRP denote RoI Transformer and the Light-Head R-CNN OBB, deformable Position Sensitive RoI pooling, respectively [8].

Image Size	LR-O	RoI Trans.	DPSRP	APP
416×416	-	-	-	13.5
1024×1024	141	170	206	140

**Table 9 sensors-20-01240-t009:** Mean length of the objects.

Object	Plane	BD	Bridge	GTF	SV	LV	Ship	TC	BC	ST	SBF	RA	Harbor	SP	HC	avg
length	1.04564	1.0169	1.7242	1.4834	2.1373	3.8274	2.9355	1.9410	1.8625	1.0174	1.3142	1.0278	4.4173	1.0914	2.9527	1.9863

**Table 10 sensors-20-01240-t010:** Angle error and pixel error.

Number	$\lambda_{noobj}$	$\lambda_{obj}$	$\lambda_{ROI}$	$\lambda_{pts}$	$\lambda_{softmax}$	Average Angle Error	Global Projection Pixel Error
1	1	2	0.0001	1	1	4.28	2.03
2	1	3	0.0001	1	1	4.32	2.05
3	1	5	0.0001	2	1	4.85	2.07
4	1	2	0.0001	1.5	1	4.93	2.16
5	1	2	0.1	1	1	4.78	2.37

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
