# Peer review of "3D Pose Estimation for Object Detection in Remote Sensing Images"

_sensors, 2020, doi:10.3390/s20051240_

Round 1

Reviewer 1 Report

Please see the attach.

Author Response

    Thank you for your review. This table is the reply to the review. Other modified parts can be found in the version with modified traces. The red ones are deleted and the blue ones are newly added. In the column Location of this table, OV indicates old version of the article; MT indicates new version with modified traces.

Number Review opinion Location Reply
1 In the Abstract section, lines
1-10, I think it is inappropriate to write this section in
the Abstract, this section is
about the problems found.
OV: lines 1-10
MT: none
As this part is repeated in the
introduction, I deleted this
part in the abstract.
2 Line 8, ”The mAP obtained
...”, Whatis mAP?
OV: line 8
MT: line 85
In line 85 of new version with modified traces, I have added the introduction of mAP. ”The mAP(meanAverage Precision), which is used to evaluate the accuracy of object detection methods, is based on IoU between
prediction boxes and ground truth boxes.”
3 In lines 11 and 12, ”Different from the previous methods” and ”Different from the traditional methods” are unclear? OV: lines 11-12
MT: line 12
Line 11 is to highlight
the differences between our
method and others. Line 12
is redundant, so I remove it.
4 In lines 11 and 16, ”Differ-
ent from the previous meth-
ods, the object results of
the final output can get the
direction information.” and
”Different from the previous
methods, the object results
of the final output of this pa-
per can get direction infor-
mation. ”, there is redun-
dancy between the two sen-
tences.
OV: line 11 line 16
MT: line 18
Line 16 is really redundant,
and I also remove it.
5 In line 31, “FPN uses feature pyramids for object detection [10];” OV: line 31
MT: line 34
Deleted the redundant vertical line.
6 In the introduction of Figure 1, some are X-axis and
some are X axis, which are
not uniform.
OV: Figure 1
MT: Figure 1
Unify X axis, Z axis into X-axis, Z-axis, respectively.
7 In line 57, ”Such as mtcnn,
Faster-RCNN, yolo, ssd”
unclear what does it mean?
OV: line 57
MT: line 60
The meaning of this sentence is that these detec-
tion methods can also be regarded as the detection of
two key points using the ideas in this article. I
changed the description in
line 60 of new version. ”The
traditional rectangle object
detection methods can be attributed to the prediction of
2 key points, such Faster-RCNN, YOLO and SSD”
8 In line 63, “the object to be
lost in the NMS operation.”,
What is NMS?
OV: line 63
MT: line 67
The full name of NMS is
non-maximum supply. This
method is used to search
the local maximum and suppress the maximum. The
purpose of this method is to
eliminate redundant frames
and find the best location for
object detection.
9 In line 102, ”We use λROI =
0.01 to assist learning”, why
set λROI to 0.01?
OV: line 102
MT: line 113
λ ROI represents the weight
of the horizontal box. ”We are more focused on the po-
sitioning learning of the inclined box determined by
APP, so we reduce the weight of thehorizontal box.
Then we use λ ROI = 0.01 to assist learning, and we
can even set it to 0 to ignore the weight of the horizontal
box.”
10 In the ”Experimental de-
tails” section, the meaning
of the red-marked representatives in the table is unclear.
OV: Table 4, 5, 8
MT: Table 4, 5, 8
In these tables, I have added some descriptions
and references. ”There 15
categories, including Base-
ball diamond (BD), Ground track field (GTF), Small vehicle (SV), Large vehicle
(LV), Tennis court (TC), Basketball court (BC), Storage tank (ST), Soccer-ball
field (SBF), Roundabout
(RA), Swimming pool (SP),
and Helicopter (HC)”...
11 In the conclusion part, there
are only advantages, not
the disadvantages of the
method.
OV: conclusion
MT: line 213
Our method does have disadvantages. ”For a slender object like Harbor, the
bounding box is relatively
large, and the object occupies only a small part of the
bounding box. In this way,the features in the extracted ROI region will be inaccurate, leading to a decrease in the accuracy of predicting key points of the object.”
This can be found in the conclusion part of new version.
12 In line 79, ”As can be seen
from the above figure, as
long as the Faster-RCNN
algorithm aligns the center
point”, I don’t see information about Faster-RCNN in
the figure.
OV: line 79
MT: line 88
Faster-RCNN refers to a neural network that can
recommend horizontal box.
Words in the original text
are easily ambiguous, so
I have replaced it with
”the neural network that
can recommend horizontal
box”.
13 In In section 4.2., “the x, y,
and z axes” should be “the x,
y, and z axis”?
OV: section 4.2
MT: section 4.2
Yes, you are right. I have changed it.

Reviewer 2 Report

This paper describes a 6D pose estimation method for remote sensing images, being this method based in ML techniques.

First of all, the manuscript must undergo a thorough language review. Please, let an english speaking native undertake this task.

My major concern with this paper is precisely this 6D, as no 6D pose estimation is computed nor, obviously, solved. It is a method to compute 2D rotation and translation, since aspect ratio, as authors state, is known. Being this the case (note that no information or mention to classic pose description as Euler angles is included in the paper), the very title of the manuscript it definitely misleading.

Putting this together with the language problems, in my opinion this paper does not reach the standards of the journal

Minor issues:

in fig 4 caption IoU App is repeated twice in fig 4, labelling includes parameters that have been not explained, and the notation IoU pts, unknown so far

Author Response

    Thanks for your review. The calculation process of the object’s 6D pose can be found in Figure 7. We have added a new method PnP to calculate the object’s 6D pose and supplementary experiments. The description of PnP method and homograph method can be found in section 3.3.1 and section 3.3.2, respectively. Predicting 6D pose of object requires 8 key points, that is, the vertices of the object's peripheral solid box. Since the distance from the camera to the ground object is much larger than the object’s own height, the image coordinates of the two feature points at different heights of the object at the same latitude and longitude are very close on the image. Our method can get X, Y, Z three axes. The X-axis of the object is marked with red, the Y-axis is marked with green, and the Z-axis is marked with blue. The X-axis points to the right side of the object; the negative direction of Y-axis indicates the static direction of the object ; the Z-axis points to the ground. From this we can get the position and attitude of the object.

    Figure 4 also has been updated. IoUpts is the IoUapp, and I'm sorry I forgot to change it here. Other modified parts can be found in the version with modified traces. The red ones are deleted and the blue ones are newly added.

    Thanks again for your review.

Round 2

Reviewer 2 Report

First of all, thank you for your efforts to improve the quality of the manuscript you have submitted. Nevertheless, regarding the statement that you're making a 6D pose estimation, I have to say that you're not right.

In order to make a 6D pose estimation you have to give, as a result describing such a pose, six parameters: (x,y,z) for position and (roll, pitch, yaw) for orientation. Any DOF under this number does not and can not describe a 6D pose.

Once this has been clearly stated, in your manuscript one can find Table 1, where you authors clearly say that for 6D pose estimation eight or more points are needed. However, you're working with four of them. As you can see, saying that you can perform 6D pose estimation is not consistent with using only 4 points.

Moreover, in your calculations using any of the two methods you introduce, you're considering that the object under analysis is attached to the base plane (Equation 6, z coordinate, line 151 and Equation 8, where simply no z coordinate is provided). Obviously, if the object is attached to the ground plane, you don't need and can not estimate three of the six parameters describing a pose in the 3D space. You cannot estimate a z, because you have supposed it is the ground plane one, i.e. z=0. You cannot estimate a pitch angle, as it would need to put the object in a plane which is not the ground one, and you cannot estimate a roll angle for the same reason.

Therefore, I have to uphold my previous opinion, in the sense that this paper is not describing a method for 6D pose estimation and, as a consequence, uphold as well my advice to you in the sense of reconsidering your paper.

Author Response

Please see the attach.

Round 3

Reviewer 2 Report

The answers and the modifications that the authors have included in the manuscript fulfill my requirements, therefore I think that the paper has got an appropriate level to be published